# Molecular bilayer graphene

Xin-Jing Zhao[1,2], Hao Hou[1,2], Xue-Ting Fan[1,2], Yu Wang[1], Yu-Min Liu[1], Chun Tang[1], Shun-He Liu [1], Peng-Peng Ding[1], Jun Cheng [1], Dong-Hai Lin[1], Cheng Wang [1], Ye Yang[1] & Yuan-Zhi Tan [1]

Bilayer graphene consists of two stacked graphene layers bound together by van der Waals interaction. As the molecular analog of bilayer graphene, molecular bilayer graphene (MBLG) can offer useful insights into the structural and functional properties of bilayer graphene. However, synthesis of MBLG, which requires discrete assembly of two graphene fragments, has proved to be challenging. Here, we show the synthesis and characterization of two structurally well-defined MBLGs, both consisting of two π−π stacked nanographene sheets. We find they have excellent stability against variation of concentration, temperature and solvents. The MBLGs show sharp absorption and emission peaks, and further time-resolved spectroscopic studies reveal drastically different lifetimes for the bright and dark Davydov states in these MBLGs.

[1] Collaborative Innovation Center of Chemistry for Energy Materials, State Key Laboratory for Physical Chemistry of Solid Surfaces, and Department of Chemistry, College of Chemistry and Chemical Engineering, Xiamen University, 361005 Xiamen, China. [2] These authors contributed equally: Xin-Jing Zhao, Hao Hou, Xue-Ting Fan. Correspondence and requests for materials should be addressed to J.C. (email: chengjun@xmu.edu.cn) or to Y.-Z.T. (email: yuanzhi_tan@xmu.edu.cn)

Single-layer graphene (SLG) possesses a variety of unique optical and electronic properties, which stem from its mono-atomic network of sp$^2$ carbons[1]. SLG sheets display an inherent tendency to stack into multilayered structures due to the inter-planar π−π interaction[2,3]. As the simplest manifestation of such structures, bilayer graphene (BLG) exhibits novel characteristics absent in SLG, such as bandgap opening[4], bound excitons[5,6], etc., and therefore provides a fertile ground[7] for studying two-dimensional materials. Advances in construction and characterization of BLG have also provided theoretical frameworks and instrumental tools in exploring the emerging field of van der Waals heterostructures[8,9]. However, current methods, such as chemical vapor deposition[10,11] and layer-by-layer assembly[12], often restrict BLG formation on substrate surfaces. Moreover, tendency of solution-dispersed graphene sheets to self-aggregation[2,3] leads to a mixture of multi-layered graphene sheets in solution and merely the dispersions of enriched bilayer or trilayer graphene sheets could be achieved by exfoliation of specific graphite intercalation compounds[13].

Nanographene is widely regarded as the molecular model of graphene[14]. Research on nanographene has provided deep insights into the structure–property relationships of graphene[15] and stimulated the field of organic synthesis of graphenic materials with a well-defined architecture[16]. Although peripheral substitution of nanographene with sterically bulky substituents can preserve its monolayer structure[17–19], stacking nanographenes in a discrete bilayer form is much more challenging. By using surface supramolecular assembly method, bilayer assembly of nanographene can only be found at liquid/solid interfaces in a very limited amount[20,21]. This problem is further compounded by the fact that the generated bilayer nanographene is often bound to, and thus difficult to be separated from, the substrate surface. An alternative strategy involves tethering two nanographene sheets together with a covalent linker[22,23]. However, the resultant product cannot be considered as a molecular cutout of bilayer graphene because the covalent attachment disrupts the van der Waals interactions between the two layers. On the other hand, although stable bilayer nanographene assembled by π−π interaction can qualify as molecular bilayer graphene (MBLG), to the best of our knowledge, such an MBLG has not been obtained in a pure form so far, because the propensity for multilayer stacking of nanographene often results in a dynamic mixture of columnar superstructures with indefinite numbers of layers[24–28].

Herein we report the synthesis and characterization of two stable MBLGs [(C$_{114}$H$_{24}$R$_6$)$_2$, **1** and (C$_{96}$H$_{24}$R$_6$)$_2$, **2**, R = mesityl], each formed by π−π stacking of two identical nanographene sheets. Mass spectrometric analysis shows that these MBLGs possess a bilayer structure and can dissociate into the corresponding monolayer when exposed to enhanced laser ablation. Nuclear magnetic resonance (NMR) spectroscopy and single-crystal X-ray diffraction (SXRD) clearly validate their bilayer structures. The interlayer H⋯H proximity in the bilayer structure is revealed by two-dimensional (2D) nuclear overhauser effect spectroscopy (NOESY). The bilayer structure of MBLGs is highly stable against varying temperature, concentration, and solvent. The absorption and emission of the MBLG shows clear vibronic fine structures. The time-resolved fluorescence (TRPL) and transient absorption (TA) spectroscopies indicate that the bright and dark Davydov states of MBLGs have drastically different lifetimes.

## Results

### Synthesis and mass spectroscopy of MBLGs.
The formation of discrete bilayer stacking requires the balance of the π−π stacking of conjugated π systems with the stereo-hindrance effects of

peripheral groups. The peripheral groups have been found to be effective for the modulation of aggregation behavior of nanographenes and a series of literature have been reported on controlling the stacking of nanographenes[29–33]. For example, the bulky mesityl group has been introduced at the peripheral to hinder interlay stacking[17–19]. On the other hand, on increasing the size of inner nanographene core, the enhanced π−π interaction can outweigh the stereo-hindrance of the mesityl group, facilitating the π−π stacking. Owing to the large size of the mesityl group[17,18,34], the π−π stacking of mesityl-functionalized nanographenes demands the interlocking of mesityl groups at both layers along the peripheral. After dimerization, the interlocked peripheral mesityl groups become congested, which could hinder further aggregation to higher oligomers. Therefore, we propose that stacking nanographenes into discrete bilayers could be achieved by making a large nanographene core functionalized with appropriate mesityl groups. Here we choose the nanographene C$_{114}$ and C$_{96}$ functionalized with six mesityl groups as the monomer of MBLGs.

Both **1** and **2** were prepared by first synthesizing the polyphenylene precursors with mesityl substituents, followed by dehydrocyclization using the Scholl reaction (Supplementary Figs. 1 and 2, Supplementary Note 1). The $m/z$ values of **1** and **2** in matrix-assisted laser desorption/ionization-time-of-flight mass spectrometry (Fig. 1 and Supplementary Fig. 3) were 4216.3 and 3784.3 Da, respectively, which precisely match the theoretical molecular mass of the corresponding bilayer structures. The experimental isotopic distributions fit the chemical formulae of **1** and **2**, respectively (Fig. 1). As the laser power increases, an additional peak appears in the spectrum of **1** at 2107.6 Da, which matches the predicted molecular weight of the monolayer fragment of **1**. Similarly, the use of more intense laser results in the generation of the monolayer fragment signal at 1891.7 Da in the spectrum of **2**. Both peaks of monolayer fragment become increasingly intense with greater laser power (Fig. 1), suggesting that **1** and **2** exist as van der Waals dimers that can dissociate into the corresponding monomers as a result of laser ablation.

### Structure of MBLGs.
The peripheral mesityl groups generally adopt a conformation perpendicular to the inner core due to the steric hindrance caused by the ortho methyl substituents[17,18,34]. When two monolayers self-assembled into the corresponding MBLG, the mirror symmetry along the monomer plane is lost. The perpendicular conformation of mesityl leads to the two ortho methyl groups being in different chemical environments, as one of them points toward (inward-facing) and the other away from (outward-facing) the interlayer spacing of MBLG (Fig. 2). These structural characteristics are clearly revealed by the $^1$H-NMR spectroscopic data. For example, the $^1$H-NMR spectrum of **1** consists of seven singlets with the intensity ratio of 2:2:1:1:3:3:3 (Fig. 2c, d). Among them, the two low-field peaks, H$_a$ and H$_b$, could be assigned to the two inequivalent peripheral hydrogen around the C$_{114}$ core. Notably, the two mid-field singlets, $H_d$ and H$_f$, correspond to the two phenyl hydrogen in the mesityl moiety and the three methyl groups are manifested by the three peaks in the high field, corresponding to H$_c$, H$_e$, and H$_g$. This is consistent with the structure of **1** described above, in which both the two phenyl hydrogen and the two ortho methyl groups in each peripheral mesityl substituent are diastereotopic due to their inward- and outward-spacing orientations (Fig. 2a, b). Not surprisingly, the predicted $^1$H-NMR spectrum (Supplementary Fig. 4) for the monolayer structure of **1** is composed of only five singlets, due to the gained mirror symmetry along the graphene plane. The same rationale could be used to explain the presence of six distinct high-field singlets in the $^1$H-NMR spectrum of **2** (H$_y$, H$_n$, H$_p$, H$_w$,

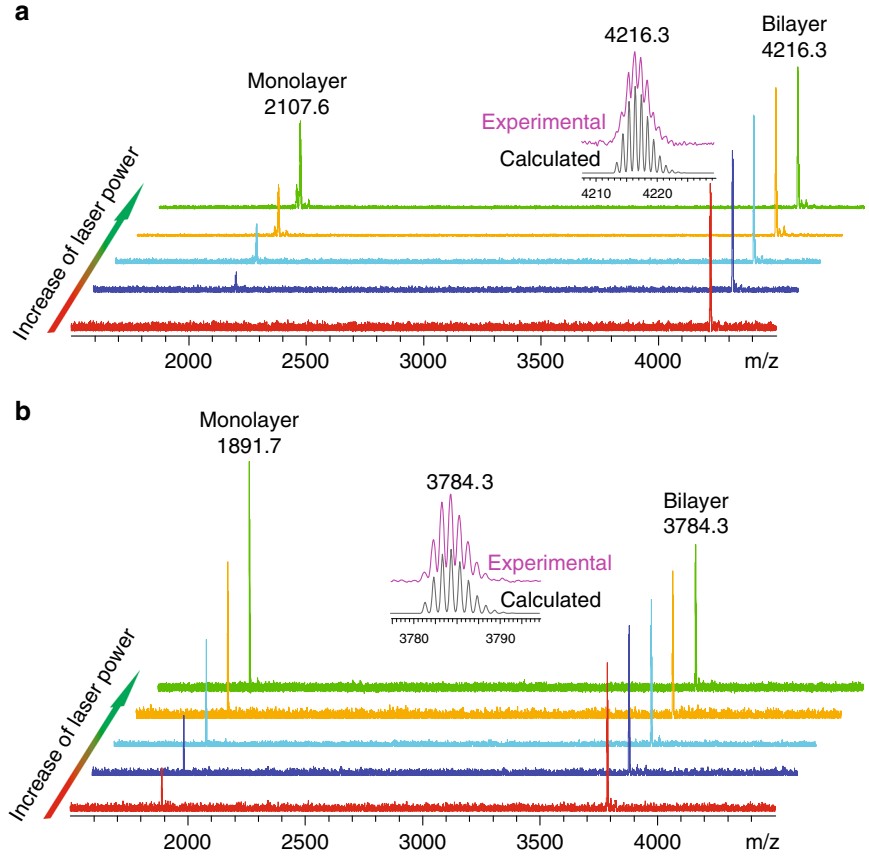

**Fig. 1** Mass spectra of molecular bilayer graphenes. The mass spectra of **1** (**a**) and **2** (**b**) with different desorption laser powder. The insert figures represent the isotopic distribution for the mass peak of **1** and **2**

$H_r$, and $H_u$), which correspond to the six methyl groups on the two inequivalent peripheral mesityl moieties (Fig. 2e–g). These assignments can be further corroborated by the $^{13}C$ NMR and $^{13}C$-$^1H$ correlation spectroscopic spectra of **1** and **2** (Supplementary Figs. 5–8).

The structures of both MLBGs are calculated by using density functional theory (DFT) (Supplementary Fig. 9) and the theoretical NMR spectra computed with the optimized structures (orange lines in Fig. 2d, h) are in good agreement with the experimental patterns (Fig. 2d, h). The presence of a bilayer structure could be further validated by the observation of 2D NOE signals attributable to interlayer proton coupling. For example, the $^1H$ signal of $H_j$ in **2** could be easily identified as the sole triplet (Figs. 2h and 3b). If considering the proton proximity within one layer only, $H_j$ is far away from the all the methyl hydrogens (>5 Å), (Fig. 3a), and therefore would not be able to couple with methyl hydrogen. In comparison, considering the bilayer structure, the inward-facing methyl hydrogen $H_n$ and $H_y$ from one layer are considerably closer to $H_j$ of another layer (Fig. 3a). Indeed, structural modeling of **2** indicates $H_j$ being 2.89 Å away from $H_n$ and 3.00 Å away from $H_y$. These distances are sufficiently close for spatial H···H coupling, to be detectable by 2D NOESY (Fig. 3b). However, $H_j$ is still too distant from the outward-facing methyl hydrogen $H_r$ and $H_u$ (7.26 and 6.71 Å). This is consistent with the fact that there are only two $H_j$-related NOE signals in Fig. 3b. The same argument could also be applied to the interpretation of the 2D NOESY of **1** (Supplementary Fig. 10).

The single crystals of both **1** and **2** can be obtained but are highly instable due to efflorescence. We modified peripheral substituents of **2** from mesityl to 2,6-dimethylphenyl and synthesized MLBG **2'** [$(C_{96}H_{24}R'_6)_2$, R' = 2,6-dimethylphenyl]

(Supplementary Fig. 11). The bilayer structure of **2'** was also confirmed by mass and NMR spectroscopies (Supplementary Fig. 11). Fortunately, the crystals of **2'** are stable enough to be measured by SXRD (Supplementary Fig. 12), which clearly discloses the bilayer structure of **2'** (Fig. 3c, d). Two cofacially stacked $C_{96}$ cores in **2'** exhibit a staggered arrangement with an interlayer distance of 3.42 Å and all the 2,6-dimethylphenyl groups adopt a perpendicular conformation as expected (Fig. 3c, d), which are in good agreement with computed structures of **1** and **2** (Supplementary Fig. 9).

**Stability of MBLG.** Generally, dimeric assemblies based on π−π interactions are often in a dynamic aggregation/disaggregation equilibrium with the corresponding monomers[35,36]. We thus investigated the stability of the MBLGs by varying the concentration, temperature, and solvent. The $^1H$ NMR (Supplementary Figs. 13 and 14), absorption (Supplementary Figs. 15 and 16), and emission (Supplementary Fig. 17) spectra of **1** and **2** all remain unchanged in the concentration range of $10^{-4}$–$10^{-7}$ mol mL$^{-1}$, suggesting that neither MBLG undergoes dissociation as a result of dilution. Similarly, the variable-temperature $^1H$ NMR spectra of **1** and **2** show no significant alteration on chemical shifts of $^1H$ signals between 10 and 120 °C, confirming that both MBLG are sufficiently stable in this temperature range (Supplementary Figs. 18 and 19). Even after adding the poor solvent, such as methanol, their absorption and emission spectra remain the same (Supplementary Fig. 20), which rules out the possibility of further π−π stacking of **1** and **2** into trimer or oligomers. The remarkable stability of the MBLGs is further supported by DFT calculations[37–43], which show a large stabilization energy

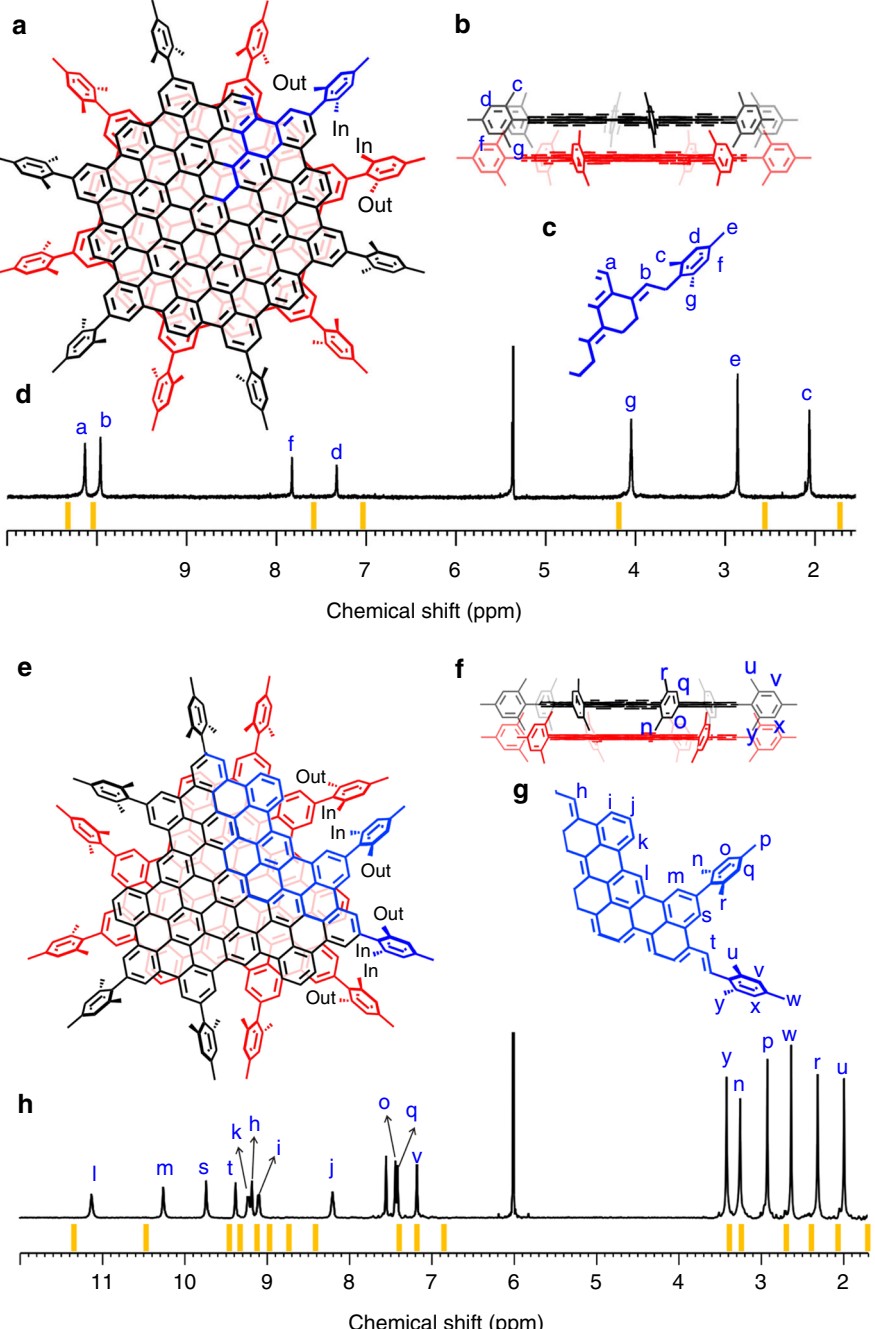

**Fig. 2** Nuclear magnetic resonance (NMR) characterization of molecular bilayer graphene. **a, e** The top views of **1** and **2**, respectively. The asymmetric units in **1** and **2** are highlighted in blue. In and out represents the inward-spacing and outward-spacing methyl of the mesityl group. **b, f** The side views of **1** and **2**. **c, g** Numeration of hydrogen atoms in the asymmetric units of **1** and **2**, respectively. **d, h** [1]H NMR spectra of **1** and **2**. The theoretical NMR spectra of **1** and **2** are represented as orange lines. All of the proton resonances are assigned with the assistance of two-dimensional NMR spectroscopy (Supplementary Figs. 21–32 and Supplementary Note 2)

(Supplementary Table 1 and Supplementary Note 3) due to the formation of the bilayer structure.

**Optical properties of MBLGs.** The photophysical properties of MBLGs were also investigated. The absorption and emission spectra of both **1** and **2** consist of a series of sharp peaks, corresponding to different vibronic lines. According to the Frank–Condon principle, the overall spectral shapes between the absorption and emission spectra exhibit an approximate mirror symmetry (Fig. 4a). The narrow line width (the same as that for single molecules) and Lorentzian line shape suggest the absence

of inhomogeneous broadening. Compared with the emission spectral shape of the monolayer counterpart with alkyl groups[28,44] (Supplementary Note 4), the first vibronic line intensity is suppressed while the second one is enhanced. Similar spectral evolution was also observed during the formation of molecular π stacks[45].

The lifetimes of the excited states are examined by the TRPL and TA spectroscopies. The TRPL kinetics monitored at the main emission peak (Fig. 4b) measures the lifetimes of both samples, which are determined to be 28.5 and 118 ns for **1** and **2**, respectively, from bi-exponential fitting (Fig. 4b). On the

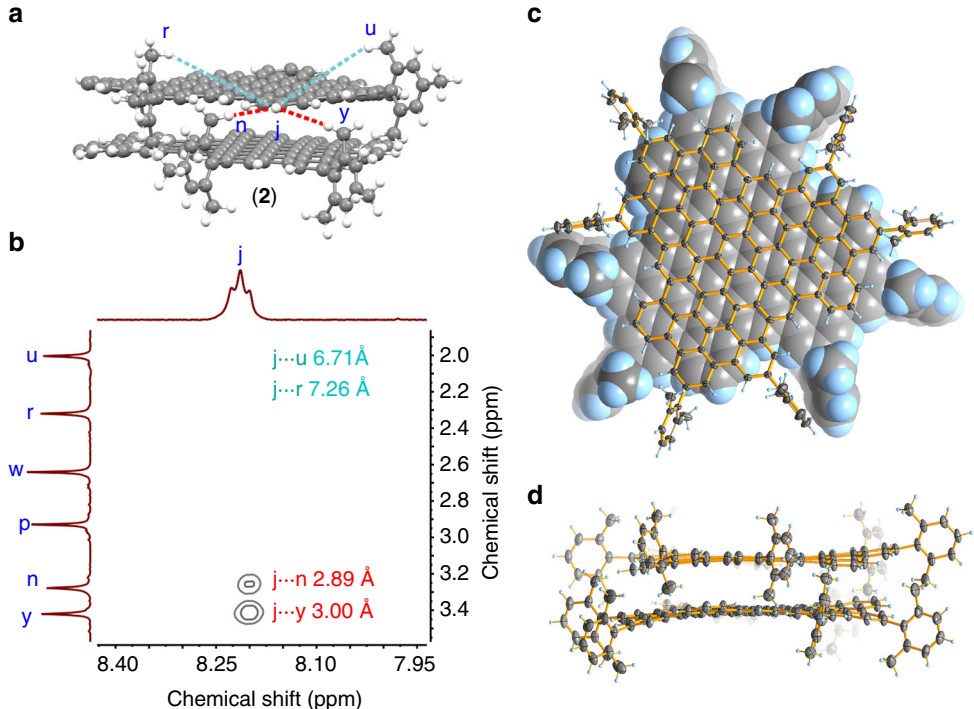

**Fig. 3** H⋯H proximity in **2** and crystal structure of **2′**. **a** Typical H⋯H proximity found in **2**. The interlayer and intralayer H⋯H proximity are represented as red and cyan dashed lines, respectively. The rest mesityl groups are omitted for clarity. **b** Expanded two-dimensional nuclear overhauser effect spectroscopy of **2**, showing interlayer proton coupling. The detailed H⋯H distances are labeled inside. **b**. **c** Top view of the crystal structure of **2′**. For clarity, one layer is illustrated with the spacefill model and the other layer with the ellipsoid model. **d** Side view of the crystal structure of **2′**. The thermal ellipsoids are set at 50% probability level

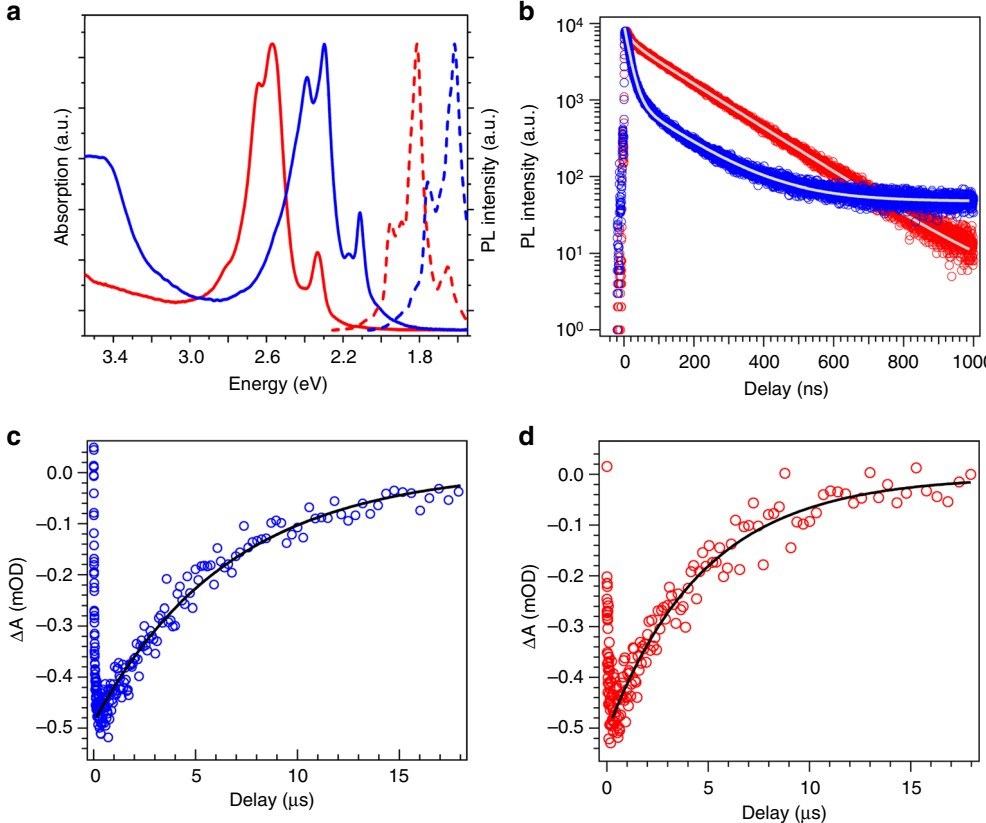

**Fig. 4** Optical properties of molecular bilayer graphenes. **a** Absorption (solid line) and emission (dashed line) spectra of **1** (blue) and **2** (red). **b** Time-resolved fluorescence kinetics of **1** (blue, monitored at 767 nm) and **2** (red, monitored at 683 nm). The average lifetime of **1** and **2** is 28.5 and 118 ns. **c**, **d** Transient absorption kinetics of **c 1** and **d 2** monitored at their respective first vibronic peak position

contrary, the TA measurements (Supplementary Note 5) indicate much longer lifetimes for both MBLGs, i.e., 5.9 and 5.0 μs for **1** and **2**, respectively, as shown in Fig. 4c, d. This discrepancy arises from the different nature of the detected states. The TRPL kinetics is mainly sensitive to the bright (emissive) state. On the other hand, the recovery kinetics of the first vibronic state bleach from the TA measurement represents the recovery rate of the ground state after optical excitation. Thus the much longer TA lifetime suggests the presence of dark (non-emissive) states that cause the TA bleach but cannot be detected by the TRPL.

Owing to the van der Waals interaction between two aggregated molecules, originally degenerate excited states from the two monomers will form excited states with different energies (Davydov splitting) after forming a dimer. The optical transition to one of these two Davydov states is allowed, while the transition to the other one is forbidden. The energetics of the allowed and forbidden states depend on the transition dipole alignment[46]. If the transition dipoles of the two monomers are in parallel alignment (such as in the case of **1** and **2**, two monomers stack facially), the upper level is the allowed state (bright state). On the other hand, if the transition dipoles are in in-line alignment, the upper level is the forbidden state (dark state). Owing to the parallel dipole alignment in π–π stacking configuration of **1** and **2**, the optical transition from ground state to the upper and lower Davydov states are allowed and forbidden, respectively[46]. Taking into account this energy splitting, we tentatively attribute the bright and dark state to the upper and lower states, respectively, and the fast energy relaxation from upper to lower states is responsible for the short lifetime of bright states.

## Discussion
We have demonstrated the chemical synthesis of well-defined MBLGs and validated their structures. The MBLGs are remarkably stable, showing no sign of aggregation or dissociation. The MBLGs here offer the well-defined molecular cutout of bilayer graphene and can help in the understanding of structure and properties of bilayer graphene at the molecular level. Further investigation on rational design of MBLGs with tunable size, edge, and twist angle could be readily envisioned, and perhaps even more ambitious goals, such as hetero-structural MBLG and bilayer graphene nanoribbons, could be aimed. Our current study can also inspire the synthesis of other stable bilayer carbonic nanomolecules, such as geodesic polyarenes, carbon nanobelt, and nanorings.

## Data availability
The X-ray crystallographic coordinates for structure reported in this study have been deposited at the Cambridge Crystallographic Data Centre (CCDC), under deposition number 1873164. These data can be obtained free of charge from The Cambridge Crystallographic Data Centre via www.ccdc.cam.ac.uk/data_request/cif.

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

## Acknowledgements

This work was financially supported by the National Natural Science Foundation of China (21771155, 21721001), the Ministry of Science and Technology of China (2017YFA0204902), the Fundamental Research Funds for the Central Universities (20720180035), and the Major Science and Technology Project between University-Industry Cooperation in Fujian Province (2016H6023). We acknowledge Zhiwei Lin for the mass spectroscopic measurements and Liubin Feng for variable-temperature NMR measurements.

## Author contributions

Y.-Z.T. conceived and designed the project; X.-J.Z., H.H., Y.-M.L., S.-H.L., and P.-P.D. conducted synthesis and completed the identification; X.-T.F., C.T., and J.C. performed the theoretical work; Y.W., C.W., and Y.Y. performed and analyzed the optical measurements. D.-H.L. analyzed the NMR data. X.-J.Z., H.H., J.C., and Y.-Z.T. co-wrote the paper; all authors discussed the results and commented on the manuscript.

## Additional information

**Competing interests:** The authors declare no competing interests.

