## [Peer Review File · Nature Communications]

Reviewers' comments:

Reviewer #1 (Remarks to the Author):

This paper reports the synthesis and optical properties of MBLGs, which refers to the pi-pi stacking of two identical well-defined graphene nanosheets. The authors demonstrate two MBLGs comprised of C114H24R6 and C96H24R6 (R is mesityl group). The MBLGs were thoroughly characterized by mass spectroscopy, NMR, NOESY and XRD, which conclusively show the bi-layer structure. In addition, the MBLGs show excellent stability under dilution, increasing temperature up to 70C, and addition of poor solvents, further confirming that the bi-layer structure is the most thermodynamically stable molecular allotrope under these experimental conditions. The concept of MBLG is very interesting because it offers a new degree of freedom to engineer single-molecule properties. In addition, the synthesis of exclusive dimer without monomer or oligomer is highly non-trivial and requires extensive efforts. Therefore, I think the paper can be potentially published in Nature Comm. However, several issues have to be clarified before this paper can be accepted.

Major points:

1. Spectroscopies show the exclusive formation of bi-layer structures, but the mechanism is still not clear. Although DFT shows very large binding energy of few eV, it does not rule out the possible formation of thicker structures, eg. trimers. The authors should perform additional DFT calculations on trimers and oligomers and compare with that of excimers. In addition, the rationale to use 1 and 2 molecules are not clear. The exclusive formation of MBLG obviously needs careful design of the molecular building blocks, so it is useful to add one discussion section about the reason to select 1 and 2, and more generally, the design rule towards MBLG, so that chemists working in this field can synthesize more building blocks.

2. The authors show the PL spectra change compared to monomers (Ref. 32) as an evidence of dimers. However, 1. The molecule in Ref. 32 was slightly different from this work; 2. Ref. 32 was measured in solid state in PS matrix, while the PL in this work is measured in solution. In order to do the correct comparison, the authors need to: 1. use the same monomer as reference, and 2. rule out the effect of different environments. They should either compare both monomer and dimer in the same solution, or in the same solid-state matrix.

3. The prolonged PL lifetime of dimers compared to monomers (Ref. 32) is not well justified. Upon the formation of dimers, charge transfer (CT) states are introduced, which acts as additional energy relaxation channel. It is expected that the lifetime of the main peak (Frenkel 0-0 transition) should decrease instead of increase. The same as my previous comments, the direct comparison with Ref. 32 may not be meaningful. Therefore the conclusion that "the fast energy relaxation from upper to lower states is responsible for the short lifetimes of bright states" is not valid.

4. Upon dimer formation, the fluorescent quantum yield normally decreases due to additional non-radiative channel (Ref. 33). Direct comparison of quantum yield under identical conditions is suggested.

Minor points:

1. In line 178, "the bright and dark state to the upper and lower states" should be "the bright and dark state to the lower and upper states"? According to Kasha's rule, the lowest emissive state is dipole allowed.

Reviewer #2 (Remarks to the Author):

In this work, the authors present the rational synthesis of two nanographenes with suitable bulky substituents and investigation on their bi-layer stacking behavior both in solution and solid-state packing. The structural analysis and optical properties are well elucidated in the context. Controlling the stacking of nanographenes has been a long-existing challenge and can be a highly attractive strategy to control the physical properties of these materials, as what has been recently widely explored for the bilayer graphene and twisting bilayer graphene by the physics community. For instance, the Müllen and Feng's groups early reported controlling the stacking of nanographenes (from the helical to the staggered superstructure with different twisting degree) in the solid-state which can result in the enhanced charge carrier transport in the discotic systems (Nature Materials. 2009, 8, 421; Chem. Mater. 2008, 20, 2872; J. Am. Chem. Soc. 2007, 129, 14116). Therefore, the current work provides another appealing and novel insight into controlling and understanding the stacking of nanographenes, with particular achievements on the synthesis of exceptional bilayer structures. I like to recommend the publication of this work in Nature Communications after addressing the following comments:

-The first paragraph of the Introduction seems not be so relevant to the main focus of current work. I would suggest the authors to describe more on the current literature efforts on the bilayer graphene and the interesting physical properties behind. Also, the authors shall introduce the previous literature efforts on controlling the stacking of nanographenes in the solid state, where accordingly the following literatures shall be cited ((Nature Materials. 2009, 8, 421; Chem. Mater. 2008, 20, 2872; J. Am. Chem. Soc. 2007, 129, 14116).

-In the section of mass spectroscopy of MBLGs, the authors may better show the spectra of higher molecular mass area (eg. >6000 m/z) in the Supplementary Information. Sometime it is common to see the MALDI-TOF mass of dimers and trimers of nanographenes in the solid stage. In this respect, it will be important to know (exclude) if there is trimers or tetramers of compounds 1 or 2 from the mass.

-For the temperature dependent NMR, did the authors measure the spectra at higher temperature than 70 degree? Normally it can be necessary to reach >120 degree in order to break up the supramolecular stacks of nanographenes in solution.

-In Figure 4a, the absorption of compound 2 is missing.

-In the Supplementary Information, the authors shall provide the full characterizations of the organic compounds, like the mass, elementary analysis, etc.

Reviewers' comments:

Reviewer #1 (Remarks to the Author):

This paper reports the synthesis and optical properties of MBLGs, which refers to the pi-pi stacking of two identical well-defined graphene nanosheets. The authors demonstrate two MBLGs comprised of C₁₁₄H₂₄R₆ and C₉₆H₂₄R₆ (R is mesityl group). The MBLGs were thoroughly characterized by mass spectroscopy, NMR, NOESY and XRD, which conclusively show the bi-layer structure. In addition, the MBLGs show excellent stability under dilution, increasing temperature up to 70C, and addition of poor solvents, further confirming that the bi-layer structure is the most thermodynamically stable molecular allotrope under these experimental conditions. The concept of MBLG is very interesting because it offers a new degree of freedom to engineer single-molecule properties. In addition, the synthesis of exclusive dimer without monomer or oligomer is highly non-trivial and requires extensive efforts. Therefore, I think the paper can be potentially published in Nature Comm. However, several issues have to be clarified before this paper can be accepted.

Major points:

Comments: 1. Spectroscopies show the exclusive formation of bi-layer structures, but the mechanism is still not clear. Although DFT shows very large binding energy of few eV, it does not rule out the possible formation of thicker structures, eg. trimers. The authors should perform additional DFT calculations on trimers and oligomers and compare with that of excimers.

Response: We thank the reviewer for this important comment. As suggested by the reviewer, we have performed additional calculations of the binding energies for the trimer formation. Results obtained by using the mixed implicit-explicit solvation approach (**Supplementary Scheme S2**) are shown in **Table 1**. We indeed find that the binding energy for the trimer formation significantly decreases compared with the binding energy for formation of the bi-layer structure. Moreover, as shown in the mass spectra up to 7000 Da, the signal of trimer has not been observed (Figure S3), suggesting unfavored formation of trimers experimentally.

Table 1 Binding energies (in eV) calculated by using the mixed implicit-explicit solvation approach with DFT-PBE functional.

	Binding Energy (eV)	
	1L+1L→2L	1L+2L→3L
MBLG 1	-4.58	-3.31
MBLG 2	-2.98	-2.09

There is however an issue that our calculated binding energies for the trimer formation are moderately negative, which seems to suggest oligomerization would be still favoured. This could be explained by two accounts. Firstly, DFT (with dispersion corrections) tends to overestimate the binding energies of aromatic systems. For example, Mackie and DiLabio calculated the aggregation of the hexa-n-hexyl-hexa-peri-hexabenzocoronene using DFT with dispersion corrections (*J. Phys. Chem. A* **2008**, *112*, 10968), and obtained a binding energy for dimer formation of -2.27 eV. While, disaggregation of the stacking structure is observed experimentally by varying concentration (*Tetrahedron Lett.* **2008**, *49*, 4869; *Org. Lett.* **2008**, *10*, 5139.). We have also tested on the dimerization energy of tri-mesityl-hexa-peri-hexabenzocoronene (tri-mesityl-HBC) (the structure is shown in **Figure 1**). Although the molecule doesn't aggregate experimentally (*Org. Lett.* **2012**, *14*, 2472), DFT calculation still gives a binding energy of -1.63 eV.

Figure 1. Structure of the tri-mesityl-HBC.

Secondly, the present solvation model may not accurately account for all important contributions to solvation free energies, i.e. entropic terms. As already discussed in our **Supplementary Information**, addition of some explicit solvent molecules in a mixed implicit-explicit solvation approach, can considerably decrease the binding energies, compared to the implicit solvation model widely used in literature. However, this approach might still not be sufficient, and lead to overestimated binding energies. Going beyond static solvation models, one would have to carry out molecular dynamics and free energy calculation (i.e. potential of mean force) on fully atomistic models (both solute and solvent molecules), which is however extremely expensive at the level of DFT. It may merit future study, but is beyond the scope of the present work.

To summarize, we have calculated the binding energies for trimer formation, as suggested by the reviewer. The mixed implicit-explicit solvation method indeed shows that there is a significant decrease in binding energies compared to dimer formation. We believe this trend predicted by DFT is reliable and is consistent with experiment, although the absolute binding energies appear to be overestimated due to the limits in DFT and solvation models used. We have replaced the **Table S1** by the Table below by including the additional calculations.

Table S1 Binding energies (in eV) in vacuum and solvent by using the DFT-PBE functional.

	Binding Energy (eV)					
	Vacuum		Implicit solvation model		Mixed solvation model	
	1L+1L→2L	1L+2L→3L	1L+1L→2L	1L+2L→3L	1L+1L→2L	1L+2L→3L
MBLG 1	-5.35	-5.92	-5.22	-6.20	-4.58	-3.31
MBLG 2	-5.41	-5.43	-5.37	-5.33	-2.98	-2.09

The corresponding description have been added at the end of **Page 34, Paragraph 4 in Supplementary Information**.

“The binding energy calculated by using the mixed solvation model for the trimer formation significantly decreases compared with the binding energy for formation of the bi-layer structure.”

Comments: In addition, the rational to use 1 and 2 molecules are not clear. The exclusive formation of MBLG obviously needs careful design of the molecular building blocks, so it is useful to add one discussion section about the reason to select 1 and 2, and more generally, the design rule towards MBLG, so that chemists working in this field can synthesize more building blocks.

Response: We appreciate the reviewer for the suggestion.

The formation of discrete bilayer stacking requires the balance of the π - π stacking of conjugated π systems with the stereo-hindrance effects of peripheral groups. The peripheral groups have been found to be effective for the modulation of aggregation

behavior of nanographenes and a series of literature have been reported on controlling the columnar stacking of nanographenes in the solid state. For example, the bulky mesityl group has been introduced at the peripheral to hinder interlay stacking. On the other hand, if increasing the size of inner nanographene-core, the enhanced π - π interaction can outweigh the stereo-hindrance of the mesityl group, facilitating the π - π stacking. Due to the large size of the mesityl group, the π - π stacking of mesityl-functionalized nanographenes demands the interlocking of mesityl groups at both layers along the peripheral. After dimerization the interlocked peripheral mesityl groups become congested, which can hinder further aggregation to higher oligomers. Therefore, we propose stacking nanographenes into discrete bilayers could be achieved by making a large nanographene core functionalized with mesityl groups of appropriate size and shape. Here, we choose the nanographene C₁₁₄ and C₉₆ functionalized with six mesityl groups as the monomer of MBLGs.

[Redacted]

[Redacted]

[Redacted]

[Redacted]

The reason to select **1** and **2** towards MBLG was discussed and added in the revised manuscript.

“The formation of discrete bilayer stacking requires the balance of the π - π stacking of conjugated π systems with the stereo-hindrance effects of peripheral groups. The peripheral groups have been found to be effective for the modulation of aggregation behavior of nanographenes and a series of literature have been reported on controlling the columnar stacking of nanographenes in the solid state. For example, the bulky mesityl group has been introduced at the peripheral to hinder interlay stacking. On the other hand, if increasing the size of inner nanographene-core, the enhanced π - π interaction can outweigh the stereo-hindrance of the mesityl group, facilitating the π - π stacking. Due to the large size of the mesityl group, the π - π stacking of mesityl-functionalized nanographenes demands the interlocking of mesityl groups at both layers along the peripheral. After dimerization the interlocked peripheral mesityl groups become congested, which can hinder further aggregation to higher oligomers. Therefore, we propose stacking nanographenes into discrete bilayers could be achieved by making a large nanographene core functionalized with mesityl groups of appropriate size and shape. Here, we choose the nanographene C_{114} and C_{96} functionalized with six mesityl groups as the monomer of MBLGs.”

2. The authors show the PL spectra change compared to monomers (Ref. 32) as an evidence of dimers. However, 1. The molecule in Ref. 32 was slightly different from this work; 2. Ref. 32 was measured in solid state in PS matrix, while the PL in this work is measured in solution. In order to do the correct comparison, the authors need to: 1. use the same monomer as reference, and 2. rule out the effect of different environments. They should either compare both monomer and dimer in the same solution, or in the same solid-state matrix.

Response: We appreciate the reviewer for the valuable comments and suggestions. We agree with the reviewer that the emission spectra of aromatic compounds depend on the molecular structure, environment and aggregation state, and direct comparison between the monomer and dimer in the same dispersion is the ideal case. However, we took effort to dissociate MBLGs into monomers by varying temperature and solvents, but failed due to the high stability of MBLGs. Thus, direct comparison between MBLG with its monomer cannot be achieved at the current stage.

Figure 5. The molecular structure of $C_{96}H_{24}(C_{12}H_{25})_6$ ($R=n-C_{12}H_{25}$) and monomer of MBLG2 (R =mesityl)

As shown in Ref. 32 (Ref.37 in the revised manuscript), a monomer photoluminescence (PL) spectrum of an analogous molecule, $C_{96}H_{24}(C_{12}H_{25})_6$, was obtained from the single molecule technique. Compared with the monomer of MBLG2, this molecule has the same aromatic core and molecular symmetry, but different peripheral groups (Figure 5). As the electronic structures of molecules are primarily determined by the aromatic core and symmetry, we can expect the similar spectral shape between these two monomers. Therefore, the single molecule PL spectrum of $C_{96}H_{24}(C_{12}H_{25})_6$ monomer reported in

Ref.32 is used as a prototype to interpret the spectral modification arising from dimerization.

Unfortunately, we are not able to use the PL spectrum of $C_{96}H_{24}(C_{12}H_{25})_6$ monomer in solution as the reference because $C_{96}H_{24}(C_{12}H_{25})_6$ ensemble in solution consists of monomers and other aggregates (*Angew. Chem. Int. Ed.* **2004**, *43*, 755). The monomer PL spectrum however could be obtained in Ref. 32 because the monomers and aggregates were immobilized in the PS matrix and the single molecule PL technique could select the monomers out of the aggregates.

Meanwhile, according to the suggestion of reviewer, we measured the PL spectra of MBLG2 in PS matrix, the same environment as reported in Ref.32. The PL spectrum of MBLG2 in PS matrix shows a nearly identical profile with that of MBLG2 in solution (Figure 6).

Figure 6. The photoluminescence spectra of MBLG 2 in solution and PS matrix.

3. The prolonged PL lifetime of dimers compared to monomers (Ref. 32) is not well justified. Upon the formation of dimers, charge transfer (CT) states are introduced, which acts as additional energy relaxation channel. It is expected that the lifetime of the main peak (Frenkel 0-0 transition) should decrease instead of increase. The same as my previous comments, the direct comparison with Ref. 32 may not be meaningful.

Response: We thank the reviewer for the comment. We agree with reviewer's comment that the direct comparison the lifetime of MBLG (The PL lifetime of MBLG2 in PS matrix was measured and determined to be 120 ns, which is very close to the lifetime in solution, 118 ns) with that of monomer trapped in PS matrix (5 ns, Ref 32) may not be as informative as we suggested, considering the lifetime could be influenced by other factors and the solution of pure monomer is not available at current stage. We therefore remove the comparison of lifetime in the revised manuscript to avoid confusion.

Therefore the conclusion that “the fast energy relaxation from upper to lower states is responsible for the short lifetimes of bright states” is not valid.

Response: The conclusion that “the fast energy relaxation from upper to lower states is responsible for the short lifetimes of bright states” was derived from the comparison of time-resolved photoluminescence (TRPL) and transient absorption (TA) analysis of MBLGs, which do not involve the comparison with the monomer. Due to the van der

Waals interaction between two aggregated molecules, originally degenerate excited states from the two monomers will form excited states with different energies (Davydov splitting) after forming a dimer. The optical transition to one of these two Davydov states is allowed, while the transition to the other one is forbidden. According to Kasha (M. Kasha et al, Pure Appl. Chem. **1965**, 11, 371-392), the energetics of the allowed and forbidden states depend on the transition dipole alignment. If the transition dipoles of the two monomers are in parallel alignment (such as in our case, two monomers stack facially), the upper level is the allowed state (bright state). On the other hand, if the transition dipoles are in in-line alignment, the upper level is the forbidden state (dark state). Therefore, the bright and dark states for MBLG in this paper correspond to the upper and lower states, respectively.

Since time-resolved photoluminescence (TRPL) measures quenching dynamics of the bright state, the lifetime extracted from TRPL is dominated by the bright state. However, either dark or bright state occupation will lead to the ground state bleach in the transient absorption (TA) spectra. Thus, the TA bleach recovery kinetics represents the relaxation from excited state to ground states. As shown in Figure 4B, C and D, the TA kinetics is significantly longer than the TRPL kinetics, suggesting that the bright state lifetime is much shorter than that of dark state.

4. Upon dimer formation, the fluorescent quantum yield normally decreases due to additional non-radiative channel (Ref. 33). Direct comparison of quantum yield under identical conditions is suggested.

Response: We agree that PL quantum yield could be varied due to the other factors rather than dimer formation. As we stated previously, a solution with pure monomers cannot be obtained at current stage. To avoid confusion, we prefer not to directly compare the quantum yield with monomer trapped in matrix, although the quantum yield of MBLG2 in PS matrix has been measured to be 8.2%, much lower than the quantum yield (35%) of monomer trapped in PS matrix (Ref.32).

Minor points:

1. In line 178, “the bright and dark state to the upper and lower states” should be “the bright and dark state to the lower and upper states”? According to Kasha’s rule, the lowest emissive state is dipole allowed.

Response: In our case (H-aggregate), the bright and dark states should correspond to the upper and lower states, respectively. (M. Kasha et al, Pure Appl. Chem. **1965**, 11, 371-392)

Reviewer #2 (Remarks to the Author):

In this work, the authors present the rational synthesis of two nanographenes with suitable bulky substituents and investigation on their bi-layer stacking behavior both in solution and solid-state packing. The structural analysis and optical properties are well elucidated in the context. Controlling the stacking of nanographenes has been a long-existing challenge and can be a highly attractive strategy to control the physical properties of these materials, as what has been recently widely explored for the bilayer graphene and twisting bilayer graphene by the physic community. For instance, the Müllen and Feng’s groups early reported controlling the stacking of nanographenes (from the helical to the staggered superstructure with different twisting degree) in the solid-state which can result in the enhanced charge carrier transport in the discotic systems (Nature

Materials. 2009, 8, 421; Chem. Mater. 2008, 20, 2872; J. Am. Chem. Soc. 2007, 129, 14116). Therefore, the current work provides another appealing and novel insight into controlling and understanding the stacking of nanographenes, with particular achievements on the synthesis of exceptional bilayer structures. I like to recommend the publication of this work in Nature Communications after addressing the following comments:

-The first paragraph of the Introduction seems not be so relevant to the main focus of current work. I would suggest the authors to describe more on the current literature efforts on the bilayer graphene and the interesting physical properties behind. Also, the authors shall introduce the previous literature efforts on controlling the stacking of nanographenes in the solid state, where accordingly the following literatures shall be cited ((Nature Materials. 2009, 8, 421; Chem. Mater. 2008, 20, 2872; J. Am. Chem. Soc. 2007, 129, 14116).

Response: Thank the reviewer very much for the suggestion. The first paragraph of the Introduction was rewritten according to the suggestion of reviewer. The description on the current literature efforts on the bilayer graphene and related physical properties was added in the reversion.

The controlling the stacking of nanographene in solid state was also discussed in the revised manuscript and related references were cited.

“The peripheral groups have been found to be effective for the modulation of aggregation behavior of nanographenes and a series of literature have been reported on controlling the columnar stacking of nanographenes in the solid state”

-In the section of mass spectroscopy of MBLGs, the authors may better show the spectra of higher molecular mass area (eg. >6000 m/z) in the Supplementary Information. Sometime it is common to see the MALDI-TOF mass of dimers and trimers of nanographenes in the solid stage. In this respect, it will be important to know (exclude) if there is trimers or tetramers of compounds 1 or 2 from the mass.

Response: Thank the reviewer very much for the suggestion. We agree with the reviewer that the MALDI-TOF mass spectra of nanographene can show the signals of dimer and trimer sometimes, but the signal intensity of dimer and trimer is much lower that of the monomer. We performed the mass spectroscopy of MBLGs with higher mass area up to 7000 m/z. The spectra were provided in the Supplementary Information in Figure S3. As shown, in the mass area up to 7000 m/z, the peak of bilayer structure is dominant and the signal of trimer (6324.5 m/z and 5676.5 m/z) can not be seen.

Figure S3 Mass spectra of **1** (a) and **2** (b).

-For the temperature dependent NMR, did the authors measure the spectra at higher temperature than 70 degree? Normally it can be necessary to reach >120 degree in order to break up the supramolecular stacks of nanographenes in solution.

Response: Thank the reviewer very much for the suggestion. The variable-temperature NMR of these MBLGs were carried out up to 120 °C and the related NMR spectra were updated and provided in Figures S18 and S19. In the NMR spectrum at 120 °C, all the peaks of ^1H are still included. The characteristic split of ^1H signal on ortho-methyl groups remains, which demonstrates the stability of MBLGs up to 120 °C. However, due to the high temperature, the rotation of mesityl group lead to the significant signal broadening for ortho-methyl groups and meta-hydrogens. In contrast, the signal for the para-methyl remains nearly intact, because rotational motion of mesityl group does not change the position of para-methyl.

Figure S18 ^1H NMR spectra of **1** in $\text{C}_2\text{D}_2\text{Cl}_4$ at different temperature.

Figure S19 ^1H NMR spectra of **2** in $\text{C}_2\text{D}_2\text{Cl}_4$ at different temperatures.

-In Figure 4a, the absorption of compound 2 is missing.

Response: Thank the reviewer for pointing this out. We have replotted the figure.

-In the Supplementary Information, the authors shall provide the full characterizations of the organic compounds, like the mass, elementary analysis, etc.

Response: The mass, elementary analysis of new organic compounds was carried out and provided in the Supplementary Information. The elementary analysis of MBLGs and their polyphenylene precursors was hindered by insufficient combustion of large aromatic hydrocarbons, which leads to unreliable elementary analysis.

Reviewers' comments:

Reviewer #1 (Remarks to the Author):

I appreciate that the authors have carefully addressed my comments. I understand that my second comment was not experimentally answered due to technical difficulties, but their argument that the PL was largely determined by the core structure was reasonably convincing. So I'm overall satisfied with the current version. I suggest that some of the key points in the reply (for examples, the reason to compare the PL with another monomer, the reason fast energy relaxation from upper to lower states is responsible for the short lifetimes of bright states, and discussions about quantum yield) should be added in the paper for the readers to better understand it.

Reviewer #2 (Remarks to the Author):

The authors have carefully addressed my concerns. Therefore this manuscript can be accepted for publication as it is.

Reviewer #3 (Remarks to the Author):

I have carefully read the revised manuscript and the response of the Authors to the criticism brought up by the Referees at the first round of refereeing. The Authors satisfactory responded to most of the Referees' questions, but there is still an issue which should be addressed before the paper can be published.

Specifically, I am concerned about the validity of the DFT calculations the Authors made. They used the empirical $1/r^6$ corrections to account for vdW interaction, while it is well known that this approach is of a very low accuracy, and frequently gives rise to qualitatively wrong results, as widely discussed in literature for stacked molecules [e.g., Nature Communications 8 (2017) 14052 DOI: 10.1038/ncomms14052; Chem. Rev. 117 (2017) 4714] or extended solids [Phys. Rev. Lett. 108 (2012) 235502]. Calculations with a real vdW XC functional are necessary to properly assess the energetics and geometry of the stacked molecules.

Moreover, the DFT results in vacuum seem to indicate that the formation of trimer (MBLG 1) is actually energetically more favorable, Table S1, left column. The binding energy is the energy released when two parts of the system are joined together, and the table reports -5.35 eV vs -5.92 eV for the $1L+1L\rightarrow 2L$ and $1L+2L\rightarrow 3L$ reactions. The implicit solvation model gives the same result, and only the mixed solvation model indicates that formation of trimers is not energetically favorable. This contradiction should also be discussed.

Reviewers' comments:

Reviewer #1 (Remarks to the Author):

I appreciate that the authors have carefully addressed my comments. I understand that my second comment was not experimentally answered due to technical difficulties, but their argument that the PL was largely determined by the core structure was reasonably convincing. So I'm overall satisfied with the current version. I suggest that some of the key points in the reply (for examples, the reason to compare the PL with another monomer, the reason fast energy relaxation from upper to lower states is responsible for the short lifetimes of bright states, and discussions about quantum yield) should be added in the paper for the readers to better understand it.

Response: We thank the reviewer for the kind considerations. As suggested by the reviewer, we have added the discussions about the reason to compare the PL with another monomer, the fast energy relaxation from upper to lower states and the quantum yield, in the manuscript and Supplementary Information.

Reviewer #2 (Remarks to the Author):

The authors have carefully addressed my concerns. Therefore this manuscript can be accepted for publication as it is.

Response: Thank the reviewer very much for the kind consideration.

Reviewer #3 (Remarks to the Author):

Comments: I have carefully read the revised manuscript and the response of the Authors to the criticism brought up by the Referees at the first round of refereeing. The Authors satisfactory responded to most of the Referees' questions, but there is still an issue which should be addressed before the paper can be published.

Specifically, I am concerned about the validity of the DFT calculations the Authors made. They used the empirical $1/r^6$ corrections to account for vdW interaction, while it is well known that this approach is of a very low accuracy, and frequently gives rise to qualitatively wrong results, as widely discussed in literature for stacked molecules [e.g., Nature Communications 8 (2017) 14052 DOI: 10.1038/ncomms14052; Chem. Rev. 117 (2017) 4714] or extended solids [Phys. Rev. Lett. 108 (2012) 235502]. Calculations with a real vdW XC functional are necessary to properly assess the energetics and geometry of the stacked molecules.

Moreover, the DFT results in vacuum seem to indicate that the formation of trimer (MBLG 1) is actually energetically more favorable, Table S1, left column. The binding energy is the energy released when two parts of the system are joined together, and the table reports -5.35 eV vs -5.92 eV for the $1L+1L\rightarrow 2L$ and $1L+2L\rightarrow 3L$ reactions. The implicit solvation model gives the same result, and only the mixed solvation model indicates that formation of trimers is not energetically favorable. This contradiction should also be discussed.

Response: We thank the reviewer for the useful comments and pointing out the relevant references on vdW corrections. Following her/his suggestion, we have performed additional calculations of binding energy for dimer formation of MBLG 1 for comparison between Grimme's dispersion correction and more advanced vdW method, namely, many-body dispersion with range-separated self-consistent screening (MBD@rsSCS). As the latter hasn't yet been implemented in the CP2K code used in this work, the additional calculations have been run using VASP. The calculations show that the optimized geometry of stacked MBLG 1 with the MBD@rsSCS method is almost the same as that

with the DFT+D3 method, and there is only a small difference of ~ 0.14 eV in the binding energy. We therefore conclude that the choice of vdW correction method won't make qualitative differences in the structures and binding energies for these molecules studied in the present work.

As the reviewer pointed out that calculations with implicit solvation show the trimerization is more favored than dimerization, we believe that this is due to the limitation of the solvation model, and that explicit solvation is key to accurately describe the aggregation of the layered nanographene molecules. As indicated in Supplementary Table 1, inclusion of some explicit solvent molecules in the model lead to the reversed trend. In vacuum, the calculation shows aggregation to higher oligomers is more favored. The qualitative result is consistent with experiment.

We have added relevant discussion in the Supplementary Information to clarify the choice of vdW correction method and the importance of the explicit solvation model in order to accurately obtain the trend in binding energies. The relevant references as mentioned have been cited in the revised manuscript.

REVIEWERS' COMMENTS:

Reviewer #3 (Remarks to the Author):

I read the response of the Authors to my comments, and although they partly addressed my concerns, I am still a bit confused. Specifically, I asked:

- (i) to check the calculations by a more accurate method;
- (ii) explain the inconsistency in the data listed in Table S1.

As for (i) the Authors say that they have done the calculations with a more accurate method, and that the geometry of one of the configurations was "almost the same as that with the DFT+D3 method, and there is only a small difference of ~ 0.14 eV in the binding energy."

However, it still not clear, if it was the lowest energy configuration (wrt other configurations), and how the energetics of bi-/tri-layers would change. This could have solved the contradiction in the data listed in Table S1.

Well, let's nevertheless assume that the Grimme method the Authors originally used gives qualitatively correct results, but then the approach itself (calculations of binding energies in vacuum) must be inadequate, as it gives the results which contradict the experiment, and only the "mixed solvation" model should be used.

Anyway, as this is not the central part of the paper, I recommend the manuscript for publication in its present form.

Response to Reviewer 3

Comments: I read the response of the Authors to my comments, and although they partly addressed my concerns, I am still a bit confused. Specifically, I asked:

- (i) to check the calculations by a more accurate method;
- (ii) explain the inconsistency in the data listed in Table S1.

As for (i) the Authors say that they have done the calculations with a more accurate method, and that the geometry of one of the configurations was “almost the same as that with the DFT+D3 method, and there is only a small difference of ~0.14 eV in the binding energy.”

However, it still not clear, if it was the lowest energy configuration (wrt other configurations), and how the energetics of bi-/tri-layers would change. This could have solved the contradiction in the data listed in Table S1.

Response: We thank the reviewer for the comments, and recommendation of the manuscript for publication. Regarding the vdw correction, our test calculations on comparing the structures of the dimer and the binding energies using DFT+D3 and the suggested MBD@rsSCS have demonstrated that these two methods give essentially the same results for the molecular graphene systems. Therefore, we believe that the MBD@rsSCS will not change the results for bi-/tri-layers as calculated by DFT+D3.

Comments: Well, let’s nevertheless assume that the Grimme method the Authors originally used gives qualitatively correct results, but then the approach itself (calculations of binding energies in vacuum) must be inadequate, as it gives the results which contradict the experiment, and only the "mixed solvation" model should be used. Anyway, as this is not the central part of the paper, I recommend the manuscript for publication in its present form.

Response: We thank the reviewer for the comments, and recommendation of the manuscript for publication. We agree that the discrepancy must result from the solvation models used. The mixed implicit-explicit solvation model gives more experimentally suitable results, which are different from the simple implicit solvation model. Following the reviewer’s suggestion, we have removed the results calculated in vacuum and implicit solvation model in Supplementary Table 1 to avoid any confusion, and accordingly revised some discussion in the Supplementary Information for clarity.